# Christian Mindfulness and Mental Health: Coping through Sacred Traditions and Embodied Awareness

Veronica L. Timbers [1,2,*] and Jennifer C. Hollenberger [1,3]

1    Diana R. Garland School of Social Work, Baylor University, Waco, TX 76798, USA;
     jennifer_hollenberg1@baylor.edu
2    School of Social Work, Boise State University, Boise, ID 83725, USA
3    Department of Psychology and Social Work, Grove City College, Grove City, PA 16127, USA
*    Correspondence: veronicatimbers@boisestate.edu

**Abstract:** Mindfulness is increasingly implemented as a tool in mental health practice for coping and self-care. Some Christians worry that these practices might be in conflict with their own tradition, while other Christian contexts are reclaiming the contemplative aspects of the faith. Though clinicians are not trained to teach on religious topics and ethically must avoid pushing religion onto clients, conceptualization and research extend the benefits of mindfulness practices for religious clients. This paper will discuss the evidence for using mindfulness in mental health treatment and connect mindfulness to the Christian tradition. The authors explore how intentional awareness and embodiment of the present moment are supported in Christian theology through the incarnation of Jesus and God's attention of the physical body in the Christian scriptures. The authors also discuss how sacraments and prayer naturally overlap with mindfulness practices for the dual purposes of emotional healing and spiritual growth. To bolster the benefits of mindfulness in the psychological and religious realms, the purpose of this paper is to empower therapists to address client concerns of whether mindfulness is in conflict with Christianity, support clients in expanding current Christian religious coping, and provide Christian leaders with more information about how mindfulness elements are already present in Christian rituals and beliefs.

**Keywords:** mental health; mindfulness; religious coping; Christian theology; Christian counseling

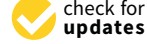



## 1. Introduction

Ask a crowd of people to define "mindfulness", and you will hear a variety of assumptions and applications. At one time, the understanding of mindfulness was inseparable from Buddhist and early Christian contexts (Knabb et al. 2019; Trammel 2017), but with more research showing the benefits of mindfulness, it has become a common word, embedded in familiar settings such as medical appointments, mental health counseling, classrooms, and corporate offices. Research on mindfulness practices shows positive evidence for improved physical health and well-being. The benefits include pain management (Jacob 2016; Monk-Turner 2003), increased cognitive well-being and flexibility (Newberg et al. 2014; Wells et al. 2013), and decreases in depression, anxiety, and psychological distress (Chen et al. 2015; Galante et al. 2014; Lee et al. 2017). In one controlled trial, a lovingkindness meditation practice was also found to decrease implicit bias and positively impact social relationships and group dynamics, showing evidence of the interpersonal benefits of mindfulness (Kang et al. 2014).

Consequently, mindfulness is increasingly implemented as a tool in mental health practice for coping and self-care; however, some Christians worry that these practices might be in conflict with their own tradition. On the other hand, recognizing the benefits of mindfulness, some Christian contexts are reclaiming and focusing on contemplative aspects of the faith, connecting mindfulness to already established religious routines and rituals (Center for Action and Contemplation 2021; Contemplative Outreach 2021; Contemplative

Society 2021). Despite this revival of interests in mystical and contemplative traditions, many Christians continue to believe mindfulness is rooted in other religious traditions, such as Buddhism, or secular humanism. Although the purpose and meaning of mindfulness is distinctive to the public or religious context where it is used (Kabat-Zinn 2003; Knabb and Vazquez 2018; Trammel 2017), the practice is based on two anchoring characteristics; intentional awareness and the embodiment of the present moment (Bishop et al. 2004; Kabat-Zinn 2013, 2003; Shapiro 2009). Shapiro (2009) has summarized mindfulness as "the awareness that arises out of intentionally attending in an open and discerning way to whatever is arising in the present moment" (p. 555). This definition opens up the application and adaption of mindfulness to many settings and religious traditions, but adaptions must be made with care to avoid misappropriation from one context to another.

## 2. Historical Contexts in Buddhism and Christianity

Though difficult to trace, it is likely that practices of meditation and mindfulness in ancient Buddhism and Judaism (the latter of which Christians connect to their tradition) began in the first millennium BC. Both traditions speak of such practices in their ancient texts but developed in unique ways based on the monotheistic and non-theistic distinctions between the two. The common belief that mindfulness is specifically a Buddhist practice is perhaps because the terms *mindfulness* and *meditation* are part of the core tenants of Buddhism. In both Jewish and Christian traditions, meditation and mindful attention are described in varied practices that are embedded within the stories of their sacred texts.

For instance, in the Buddhist context, *right mindfulness* and *right concentration* are two of the steps in the Eightfold Path that guide the practice of Buddhism (Prothero 2010; Smith 1991). Right mindfulness is defined as deep, non-reactive self-awareness allowing for conscious presence and freedom from the suffering brought on by self-interest (Prothero 2010; Smith 1991). Right concentration is the explicit practice of meditation as a tool in reaching Enlightenment (McMahan and Braun 2017; Smith 1991). As one moves toward Enlightenment, they increase self-awareness about one's desires, fears, bodily sensations, and emotional attachments so that they can accept and let go of thoughts and feelings that anchor oneself to the wavering and unpredictable realities of one's self-conscious states (McMahan and Braun 2017). This concentration, then, helps one move to a higher level of consciousness which cultivates an absence of selfishness and reduction in suffering (McMahan and Braun 2017). Though different branches of Buddhism practice mindfulness and meditation slightly differently, the goal remains detachment from self and suffering.

The Christian context also emphasizes transcending self-interest and suffering; however, in Christian mindfulness, these goals are not achieved through detachment but through intentional and embodied attachment to God and others. In this tradition, God is an ever-present *object* that connects and cares for all of life and through whom suffering is relieved. In Christian practices, this may include sitting with and embracing the mystery and unknowability of God (*via negativa*). Other times, this practice is directed toward contemplating specific truths of God (*via positiva*) (Trammel 2017). Kopel and Habermas (2019) describe via *positiva* practices as the mind being "occupied by contemplating a truth, such as key Scripture passages, God's attributes, God's promises, or even the reality of heaven (p. 309). For instance, the Book of Psalms, which is part of the early Jewish and, later, Christian tradition, has many hymns noting the benefits of embodied worship and spiritual reflection on God. Psalms 1:2 says that those who meditate on the law of God "day and night" will be blessed. Psalm 63: 5–6 says "My soul will be satisfied as with fat and rich food, and my mouth will praise you with joyful lips, when I remember you upon my bed, and meditate on you in the watches of the night." Similar patterns of embodied awareness toward God and others are noted in the Book of Acts which describes the Early Church, as devoted to prayer, teaching, eating, and fellowship with one another (Acts 2:42, Acts 4:31–35; Acts 20:7–12).

This emphasis on intentional and embodied reflection on God continued through the Catholic monastic life and through Christian mystics, also known as the Desert Mothers and Fathers, who are connected to the early Eastern Orthodox traditions (Paintner 2012; Trammel 2017). In the early 5th century, John Cassian described the monk's purpose as "total and uninterrupted dedication to prayer" (Cassian 1985, p. 101), while Paintner (2012) says that the Desert Mothers and Fathers emphasized the importance of sitting with oneself and one's emotions in order to encounter God deep in one's soul. Today, a new monasticism has evolved where groups of Christians commit to living in community and dedicate themselves to contemplative reflection on God, neighbor, and creation (Andersen 2007).

Although the categories of Christians mentioned above are often viewed as having a unique calling to the religious life, all Christians are to dedicate themselves to intentional awareness of God, in their heart, soul, mind, and strength (Deut. 6:4; Luke 10:27; Mark 12:30). Church services, retreats, and religious holidays evidence the importance of dedicated times of reflection on God that can be informed by explicit connections to mindfulness in which a spirit or open curiosity and awareness of the moment are highlighted. Furthermore, the liturgical calendar marks the sacred rhythms of the church where specified colors are displayed, known scriptures are read, and members engage in seasonal acts of reverence to increase intentional awareness on an aspect of God or God's relationship to humanity. These known rhythms provide an opportunity for present, embodied awareness and attention to that which is beyond oneself, that is characteristic of mindfulness practices transcending the daily *noise* or chaos of life.

By connecting the core concepts of mindfulness to Christian theologies and rituals, therapists, pastoral care providers, and Christian leaders can address concerns that mindfulness is rooted to other religious traditions and support Christian believers in expanding their current Christian religious coping practices. For the purpose of this paper, positive religious coping is defined as using one's religious beliefs and practices to improve adaptive functioning or initiate transformative change by shifting meaning-making, perceptions of control, motivation for change, level of comfort, or one's sense of connection (Gall and Guirguis-Younger 2013; Lehmann and Steele 2020; Pargament et al. 1998). The authors provide an overview of how mindfulness is currently conceptualized and used in mental health treatment. Then, they present evidence for the use of Christian mindfulness practices in mental health treatment and connect mindfulness to Christian theology and traditions. By exploring the incarnation of Jesus, the theme of God's care of the physical body in the Christian scriptures, and ritual acts of sacraments and prayer, the authors ground mindfulness practice in the Christian religious context so that therapists and religious leaders can help Christian followers connect their beliefs to the emotional, physical, and spiritual benefits of the practice.

## 3. An Overview of Mindfulness in Mental Health Treatment

### 3.1. Third Wave Behavioral Interventions

Mindfulness is a practice that has been incorporated into what is referred to as the third-wave of behavioral theory (Hofmann and Asmundson 2008; Linardon et al. 2017; Ost 2008; Trammel 2018). These third-wave therapies build on cognitive-behavioral theory but also distinctively pivot in a new direction that emphasizes regulating emotions after they have been activated rather than preventing emotions from arising (Hofmann and Asmundson 2008; Linardon et al. 2017; Ost 2008). Moving away from a focus on symptomology only, third-wave behavioral approaches focus on holistic well-being related to the context and function of behavior through mindfulness, acceptance, non-judgment, self-compassion, and emotional awareness (Hofmann and Asmundson 2008; Linardon et al. 2017; Ost 2008). Because the focus is on observing the mind-body processes in a given moment, these approaches naturally incorporate metacognition, values, and spirituality which were often left out of traditional CBT methods. Linardon et al. (2017) provide an example of the use of mindfulness-based interventions in the treatment of eating disorders saying,

"Meditation, body scans, and mindfulness exercises are utilized. Present moment awareness is one of the purported mechanisms of action. Here, individuals are encouraged to relate to one's thoughts merely as events passing by, allow[ing] individuals to interpret thoughts and feelings in a non-judgemental fashion without the need to resort to avoidance or escape-related (e.g., purging) behaviours." (p. 133)

In Dialectical Behavioral Therapy (DBT), clients learn skills to positively change their behavior responses through non-judgmental observations of their self and their relation to others, thus increasing self-awareness (Dimeff and Linehan 2001; Jennings and Apsche 2014). Two of these specific skills include radical acceptance and attention to the present moment, which are rooted in Zen Buddhism and Western contemplative practices (Dimeff and Linehan 2001). These contemplative practices crossover significantly with Eastern Orthodox, Christian mystic, and monastic traditions (Trammel 2018). Additionally, Dimeff and Linehan (2001), the clinicians who developed DBT, note that once clients can moderate their emotional reactivity, the goal of treatment is to increase their overall sense of joy and transcend emotional and interpersonal dysregulation (Dimeff and Linehan 2001; Jennings and Apsche 2014). Joy and transcendence are typically words based in religious and spiritual discussions, evidencing that these therapies are complimentary of religion and spirituality. These religious connections through the lens of Christian beliefs and traditions will be discussed below, but first, it is important to show the positive outcomes of these third-wave therapies.

### *3.2. Outcomes and Evidence of Mindfulness Therapies*

Mindfulness-based interventions show positive outcomes for those who report a wide range of symptoms of trauma and other mental health disorders; therefore, the use of these interventions has become a standard in mental health practice. Additionally, the use of mindfulness practices adapted specifically to Christian beliefs is an emerging area of research (Ford and Garzon 2017; Frederick and White 2015; Knabb et al. 2019; Trammel 2018). Consequently, mental health clinicians should consider integrating religious-adapted mindfulness practices with clients who name the Christian faith as an important part of their development of coping skills and their value-system. Below is a brief overview of the general benefits of mindfulness therapies in non-religious settings as well as evidence specific to Christian mindfulness mental health interventions.

### 3.2.1. Evidence for Mindfulness Interventions in Non-Religious Therapy

In a meta-analysis of five randomized controlled trials, researchers found that among clients with a diagnosis of Borderline Personality Disorder, DBT significantly reduced suicidal acts and self-harming behavior compared to those in treatment as usual (TAU) (Hedges' g = 0.622, medium effect size) (Panos et al. 2014). DBT also showed slight improvement in patient compliance, which was evidenced by a reduced attrition during treatment (Hedge's g = 0.168). In another randomized trial, Possemato et al. (2016) used a brief four-session mindfulness intervention with veterans diagnosed with Post-Traumatic Stress Disorder (PTSD). Veterans that completed the four sessions reported decreases in PTSD and depression symptoms, which was supported by statistically significant findings between those receiving the mindfulness intervention and those receiving TAU. Williams et al. (2014) looked at mindfulness-based cognitive therapy (MBCT) in a randomized trial comparing MBCT client outcomes with clients receiving treatment as usual and those receiving cognitive psychological education. The research team found evidence that MBCT was statistically significant in reducing relapse of depressive symptoms for participants with a history of childhood trauma (Williams et al. 2014). Finally, a systematic review of third-wave therapies, by Linardon et al. (2017), revealed clients in a mindfulness treatment group demonstrated statistically significant improvements in clients with eating disorders during and post-treatment when compared to wait-listed clients. Moderate improvements were also noted in depression and self-esteem scores, suggesting benefits beyond symptom

management. The findings here are only a small sample of the overwhelmingly positive evidence for using mindfulness practices in the treatment of mental health, but it illustrates mindfulness practices are beneficial to a wide range of clients with diverse symptomologies. Now, it is also important to look at the growing body of research that considers the integration of mindfulness and religious practice.

### 3.2.2. Evidence for Mindfulness Interventions in Christian-Adapted Mental Health Therapy

In mental health research, there is a growing awareness and interest in adapting interventions based on race, ethnicity, culture, gender, religion, and intersectionalities of these identities. Researchers are explicitly acknowledging that client experiences and worldviews need to shape therapeutic approaches. Several researchers have designed randomized controlled trials for Christian mindfulness interventions showing benefits to integrating religious traditions in these practices. In a 2018 randomized controlled trial of a contemplative prayer program, Knabb and Vazquez (2018) found that a 2 week contemplative program for Christians resulted in reductions in perceived stress and an increased sense of surrender to God. Surrender to God is defined as "yield[ing] to God in order to change their evaluation of environmental demands" (p. 40). This is similar to the openness and non-judgmental stance that are highlighted in definitions of mindfulness mental health interventions. The 2 week program was based on using the Jesus Prayer, a centuries-old prayer often used in contemplation ("Lord Jesus, Son of God, have mercy on me"), to support clients in centering one's attention to God. One thing to note is that while the Jesus Prayer is widely known across different Christian traditions, it is not frequently used by all traditions. Some clients might not be familiar or comfortable with this practice without more education or resources.

Another important implication that Knabb and Vazquez (2018) note in their study is the difference in the Christian and Buddhist adaptions of mindfulness. They emphasize, that in Christian mindfulness, prayer draws one's attention to God, whereas in Buddhist practice, mediation focused on non-judgmental awareness that transcends attachment to pleasure or pain (Knabb et al. 2019). Ford and Garzon (2017) also note this difference of attention in Buddhist-influenced mindfulness versus Christian mindfulness, noting that Christian-adapted mindfulness practices focus attention and present awareness on God's presence within the moment. In a study that compared a conventional mindfulness program with a Christian-adapted mindfulness program among students, staff, faculty, and partners of these groups, Ford and Garzon (2017) found that both programs reduced stress; however, the group using the Christian mindfulness program reported higher overall reductions in psychological distress on the Depression Anxiety Stress Scale.

Finally, moving beyond just mindfulness as it relates to prayer, a study by Trammel (2018) used MP3 audio recordings of Christian-based mindfulness practices that integrated elements of centering prayer, Lectio Divina (a contemplative approach to reading scripture), and guided imagery focusing on sacred images. Trammel (2018) found similar reductions in perceived stress as the other studies noted here. The experimental group reported reductions in their perceived stress and increased scores on the Mindful Attention and Awareness Scale (Trammel 2018). These findings not only continue to support the benefits of mindfulness but support the use of interpreting and adapting established religious traditions to these types of interventions.

Although the evidence in this section shows how mindfulness is being adapted to Christian practices related to prayer and contemplative spiritual disciplines, there are still few resources for how mental health clinicians and Christian leaders can link sacraments, rituals, and the tenants of Christian faith to mindfulness. As noted above, some of these Christian contemplative practices, are being renewed in Western Christian contexts but not all Christian traditions will be familiar with centering prayer, the Jesus Prayer, or Lectio Divina. Moreover, Christian clients who are familiar with contemplative practices might

not know how these practices overlap with the core elements of mindfulness or can be used as a resource for overall well-being.

Mental health clinicians would benefit from having some knowledge of these adapted practices. They would also benefit from knowing the religious roots of mindfulness in order to help support clients who are interested in using their Christian beliefs and practices as coping tools but who also feel uncomfortable with assumptions about mindfulness and meditation being borrowed from other religions (Ford and Garzon 2017; Knabb et al. 2019; Trammel 2017). Additionally, Christian leaders can benefit from more education on how the core elements of mindfulness are based in Christian theology and already embedded in sacraments and rituals, for the sake of helping parishioners tap into the wider benefits of Christian traditions to reduce perceived stress, depression, and anxiety.

### 4. Mindfulness through the Lens of the Incarnation

The Christian belief in a Trinitarian God means that God is embodied through the incarnation of Jesus Christ, the Son, who is both fully human and fully God. Incarnation is when God, a spirit, or a conceptual quality transitions from abstract form to tangible experience. Immediately, there is a connection to the elements of mindfulness in God's intentional embodiment of a particular moment in human history when Godself entered into humanity through the birth, life, and death of Jesus. In the opening chapter of the Book of Matthew, Jesus is called "Immanuel", meaning "God with us" (1:22–23, ESV). The writer of the Book of John calls Jesus "the Word" saying "the Word became flesh and dwelt among us, and we have seen his glory, glory as of the only Son from the Father, full of grace and truth"(John 1:4, ESV). In the homily of Hebrews, the writer points to the importance of the incarnation in how Jesus is able to mediate on humanity's behalf by having lived the human experience, saying

> "Therefore [Jesus] had to be made like his brothers[humanity] in every respect, so that he might become a merciful and faithful high priest in the service of God, to make propitiation for the sins of the people. For because he himself has suffered when tempted, he is able to help those who are being tempted." (Hebrew 2:17–18, ESV)

The oft-repeated description of Jesus' nature being fully God and fully human was decided at the First Council of Nicea in 325 AD to settle debates around how Jesus' humanness and divinity fit within the monotheist view of God (González 2010). The council affirmed Jesus' nature through the term "homoousios" which means "same substance" (González 2010). Jesus is one substance with God and with humanity. Theologian Frank Senn (2016), writing on embodied theology, says that it is not just the incarnation of the Word becoming flesh but also Jesus' ascension that unites God with human experience. Senn (2016) says, "[B]ecause Christ is also the Son of God and the second person of the Trinity, his body has been taken into the Godhead. In Christ, therefore, God has a body" (p. 25).

The doctrine of the Trinity and Jesus' incarnation does not deny the physical self; instead, it affirms the realities of the human experience and points to God's knowing of these experiences. As such, Christians can bring overwhelming physical and emotional experiences to God through mindfulness techniques. For example, as one intentionally becomes aware of a specific embodied experience they can combine this self-awareness with attention to God, who cares for and ministers to each unique person. Therefore, the use of mindfulness techniques like body scans, breathing exercises, and self-soothing techniques do not conflict, but align with and add to Christian worship and surrender to God. Additionally, tending to one's mental health needs can further spiritual growth by freeing up energy that was previously used to manage emotional or physical suffering.

Finally, it should be noted that, despite the benefits of mindfulness, Christian theology would not assert that self-awareness and emotional regulation are the end goal. These practices are tools for deepening religious meaning and commitment. These practices augment religious coping and worship. In this way, the mind (the knowledge of Christian

tenants and God's character) is integrated with the physical experience (the bio-psycho-social realities) allowing for faith to be a holistic experience. The next section expands on this aspect, pointing to scriptures in which God uses physical and sensory experiences to reveal more Godself and God's loving care.

## 5. Affirmation of God's Care for the Physical Body in Scripture

Throughout the Old and New Testaments, it is clear that God validates humanity's needs, limits, and strengths, using mind-body experiences as tools for spiritual growth and social redemption. Even in the creation story, God gives humanity food, water, light, and ecosystems; the basic needs for humanity to thrive physically (Genesis 1). In this early creation, God also answers the emotional need of intimacy and companionship by creating a partner to Adam. In Genesis 3, God acts to address humanity's nakedness, or vulnerability, that develops out of a shame reaction to the first sin. From the beginning of scripture, God shows an intentional care for the mind-body-spirit experiences of humanity.

In scripture, people are not just cared for but also respond to God with their mind, body, and spirit. Passages are full of examples of people singing, kneeling, fasting, and dancing as ways to honor God. For instance, Psalm 96 says,

"Oh sing to the LORD a new song;

sing to the LORD, all the earth!

Sing to the LORD, bless his name;

tell of his salvation from day to day.

Declare his glory among the nations,

his marvelous works among all the peoples!" (v 1–3, ESV)

In 2 Samuel 6:14, David, who returned the Arc to the Jewish people, "danced before the LORD with all his might, wearing a priestly garment. So David and all the people of Israel brought up the Ark of the LORD with shouts of joy and the blowing of rams' horns" (ESV). In the Book of Esther, Esther's actions and the people's fasting give way to God rescuing the Hebrew people, which is followed by celebratory worship. God commands that during this celebration, gifts of food are to be shared, with a specific note that the poor are also to be given food (Esther 9). Even as the Jewish people worship God with their whole body during this celebration, God affirms the meaning, the physical realities, and the social quality of the worship. On the other hand, when the Hebrew people abandon God's call to "seek justice, correct oppression and bring justice to the fatherless" (Isaiah 1:17, ESV), God calls these embodied offerings and rituals empty of meaning and a "burden" to Godself (Isaiah 1:13–16, ESV). Without right meaning and purpose, the physical is simply performance. Consequently, it can be said that Christian worship is holistic, aligning the purpose of worship with the physical, emotional, and social awareness of self to God. These scriptures also affirm that intention and attention to one's mind-body-spirit can be a revelatory and meaningful act of worship, linking it naturally with the concepts fundamental to mindfulness.

Finally, in the New Testament, the story of the Samaritan woman at the well (John 4) provides an important example that merges the incarnational theology discussed above with the scriptural theme of God's holistic care for the mind, body, and spirit. The story notes that Jesus is "wearied" (v 6) after walking through the desert. He is familiar to the human experience of thirst and breaks a social boundary by asking a foreign woman for a drink. As Jesus talks with the woman, he relates the sensory experience of thirst to the fulfillment of spiritual and emotional needs through connection with God saying,

"Everyone who drinks of this water will be thirsty again, but whoever drinks of the water that I will give him will never be thirsty again. The water that I will give him will become in him a spring of water welling up to eternal life." (John 4:13–14, ESV)

Jesus is not literally saying the body will no longer require thirst but is revealing, to this woman who lives in a desert, something about God's nature through a physical sensation she understands. In this way, the scripture embraces physical experiences as a path to greater self and spiritual awareness. Moreover, during this exchange, Jesus acknowledges the woman's experiences of abandonment, marginalization, loneliness, and shame, which are experiences that can produce strong emotions similar to those addressed by clients in mental health treatment today. Though Jesus obviously does not offer the woman Christian-adapted mindfulness techniques, the passage points to the importance God places on physical, emotional, and social needs, as well as how connection with God can ease those needs. Though this section is brief in its scriptural review, it supports and hones the concepts of Christian mindfulness practice while also opening a conversation for scripture to consider the mind, body, and spirit connection more deeply as it relates to religious coping.

### 6. Sacraments and Rituals

Anthropological definitions of religion consistently note the importance of rituals in maintaining social and individual commitment to one's tradition (Kottak 2011). The reason for the importance of rituals is that they have a transcendent quality, giving meaning to common postures and activities, that increase connection to the Divine and connection to other's practicing the tradition. Within religious scholarship, sacraments are defined as an external symbol or sign of an invisible reality (Senn 2016). In Christianity, the symbol or sign is often further defined as a means of grace, or in other words, one of the ways Christians have to connect with God in the present moment (Senn 2016). The power of ritual is not only the meaning but the sensory experiences that accompany it.

For instance, in the Christian tradition, there are nuanced understandings of what communion means and if the bread and body are a symbol or actual presentation of Christ's body and blood. Despite the varied interpretations, several aspects hold true. Communion typically takes place communally and the leader will tell the story of Jesus serving the same meal to his disciples. Some traditions have collective or private time for confession and prayer. To receive the elements of communion, one may stand in line or have a fellow worshipper pass a tray. A piece of bread or cracker is tasted with wine or juice. The textures and sometimes smells of the elements are physically experienced. Some may kneel during or before this event. Often there is music for listening or singing. In this one ritual event, all five senses are activated and attuned toward God, self, and community. Though this event is not intended as a mindfulness practice, it has many of the elements of mindfulness with the intentionality, spiritual openness, and the embodiment of the present moment being rooted within the ritual.

Senn (2016) notes the similar sensory experience of baptism. He says,

> "God is acting on our bodies in the sacrament of Holy Baptism no matter how it is performed, some part of the body gets wet, even if only the head, and oil may be applied, again perhaps only on the forehead in the shape of a cross." (Senn 2016, p. 68)

During this event, there may be specific clothing, postures, singing, prayer, and potentially the laying of hands depending on the specific Christian tradition. The sensory elements unite meaning with the bodily experience of the ritual.

Clinicians can encourage Christian clients to reflect on the ritual experience to help inform and connect to other mindfulness practices. Using Socratic questioning and reflective statements, skills common in mental health dialogue, clinicians can help clients link the stillness and the presence of mind that they have experienced in worship to help them understand the intentionality and awareness used in meditation, guided imagery, or grounding techniques. Clinicians can also assist clients with merging important ideas from rituals with Christian-adapted mindfulness practices, such as using water sounds with a guided imagery that might connect to the ideas of renewal in baptism. Finally, religious leaders can tap into the benefits of mindfulness in this natural overlap between ritual and

mindfulness practice by encouraging parishioners to purposefully turn their attention to God and listen to their body and spirit in the sensory experiences of the ritual. Parishioners can be encouraged to notice the whole-bodied experience of the worship without judgment, end-goals, or a need to perform.

## 7. Prayer

The last connection to mindfulness that must be made is prayer. As the research above noted, prayer is easily linked to mindfulness, but it is important to note that not all prayer practices include the core elements of mindfulness. Even though all prayer turns one's attention to God, kataphatic prayer is action oriented using words, music, symbols, and ideas to commune with God, leaving little space for open awareness and attention to the present moment (Knabb et al. 2019). Mindful prayer is often referred to as apophatic prayer which includes centering prayer and the contemplative practice of sacred waiting. Apophatic practices are devoid of almost all content as the practitioner ceases doing and turns to conscious abiding in the presence of God (Knabb et al. 2019). Because these practices are devoid of goal-oriented striving, mental health clinicians must take care not to use apophatic Christian practices for only psychological benefits.

### 7.1. Centering Prayer

Centering prayer is a Christian practice that arose from Eastern Orthodox traditions, early Christian mystics, and monastic communities (Johnson 2018; Knabb et al. 2019; Trammel 2017). It was revived in modern Western practice by Catholic theologians, such as Thomas Keating, who describes centering prayer as "a movement beyond conversation with Christ to communion with him" (2016, para. 2). Centering prayer, then, is time spent in the presence of God that is absent of supplications, performance, or goal-oriented purposes (Johnson 2018; Keating 2016; Knabb et al. 2019). Sometimes a word, such as a name for God, or a word that relates to God, such as "love" or "peace", is used to recenter one's attention if the mind wanders; however, the practice is not to be driven by words (Keating 2016). Keating (2016) is clear that centering prayer is not a relaxation technique or for psychological purposes, though he notes people may experience positive psychological effects as a result. Centering prayer is, instead, a spiritual practice based in faith and selfless love. For this reason, centering prayer should not be recommended by mental health clinicians as simply a religious version of a mindfulness tool. Centering prayer should be rooted in its spiritual purposes and be integrated into wider mindfulness work so as not to shift the practice toward productive purposes focused on self over God.

### 7.2. Sacred Waiting

Sacred waiting is a less defined practice but is often discussed in the writings of contemplative Christians. In the book, *When the Heart Waits*, writer and theologian Sue Monk Kidd (1990) says,

> "I had to face the fact that my inability to wait was symptomatic of something amiss in my soul. I feared waiting because such pauses in life brought me close to the dark holes and empty pockets inside me, to the rigidities and self-lies, I had fashioned." (p. 31)

She goes on to discuss that the struggle to pause is a result of social messaging that worth is tied to productivity and busyness becomes an easy distraction from pain. The quieting of oneself to wait is difficult because of what many practitioners call "monkey mind" (Vago and Zeidan 2016, p. 97). Monkey mind is a metaphor describing how the mind rushes from idea to idea or worry to worry, similar to how a monkey moves swiftly among tree branches (Vago and Zeidan 2016).

The scriptural basis for waiting and quieting oneself before God is set forth in Romans 8:23–27. The passage states that, when one does not know or have the words to pray, they can rest in the presence of the Spirit praying over them. The text reads,

"And not only the creation, but we ourselves, who have the first fruits of the Spirit, groan inwardly as we wait eagerly for adoption as sons, and the redemption of our bodies. For in this hope we were saved. Now hope that is seen is not hope. For who hopes for what he sees? But if we hope for what we do not see, *we wait for it with patience.* Likewise the Spirit helps us in our weakness. *For we do not know what to pray for, as we ought, but the Spirit himself intercedes for us with groanings too deep for words.* And he who searches hearts knows what is the mind of the Spirit, because the Spirit intercedes for the saints according to the will of God." (Rom. 8:23–27, ESV, italics added)

Sacred waiting, like centering prayer, is intentionally focused on God and avoids filling the time with words, but there is often a more goal-directed purpose in waiting. An individual might seek comfort, knowing one's deepest needs, vulnerabilities, affections, and appreciation are known without words. It can also be a time of silent surrender, humbly acknowledging to God a question that is unanswered or an event that cannot be changed. Waiting is an act of holding vigil for oneself and others in the presence of God. It can also be a holding place for grief. Theologian Richard Rohr (2009) describes waiting as a way to live with the paradoxes of the world, letting go of attachment to certainty. The sacred waiting is a sitting with tension and ambiguity in the presence of an all-knowing God.

Theologically, this waiting with God and the role of the Spirit mediating on the believer's behalf, relates to the concept of perichoresis. This is a Greek term that means to the circle around (Stamatović 2016). It is used to describe the holy intimacy, oneness, and mutual indwelling of the three persons of the Trinity (Stamatović 2016). When waiting with God and allowing the Spirit to intercede, the believer is welcomed into this holy intimacy, being circle around and ministered to by God who is both three in one and one in three. This practice of sacred waiting is closely related to religious surrender and the psychological-based mindfulness practice of radical acceptance that was discussed in the above research (Linehan and Wilks 2015; Skerven et al. 2019). Clinicians can link sacred waiting to elements of mindfulness, providing a Christian lens or adaptions to non-religious mindfulness practices.

## 8. Conclusions and Implications

The purpose of this paper was to present a Christian lens to mindfulness practices so mental health therapists and religious leaders better understand how to support these practices with Christian clients. Though therapists are not trained to teach on religious topics and ethically must avoid pushing religion or spirituality on to clients, there is a responsibility for therapists to provide culturally informed approaches in therapy, including incorporating clients' religious beliefs. Based on the research above, there are several recommendations for how therapist can engage mindfulness practices.

First, using open-ended questions, therapists can allow clients space to talk about meaningful religious practices and important rituals. A therapist could begin with a statement such as, "Tell me about some of your religious practices that are most important to you." Another option is to adapt one of a Cultural Formation Interview questions from the *Diagnostic and Statistical Manual of Mental Disorders-Fifth Edition* (American Psychiatric Association 2013) asking, "Often, people look for help from many different sources, including different kinds of doctors, helpers, or religious sources. In the past, what kinds of help, advice, or practices have you sought for your [problem]?" These types of statements and questions could naturally bring to light topics of prayer, journaling, and other religious activities that elicit comfort. Then, a series of follow up questions focusing on how the client feels emotionally *and* physically before, during, and after these practices, can build client self-awareness around the existing benefits that can be expanded upon. Through psycho-education, therapists can link Christian practices and traditions to positive aspects of mindfulness, in order to support the intentionality of how the client uses the practices.

For clients, more concerned with the roots of mindfulness being in opposition to Christian tradition, therapists can select easy to understand excerpts from research (such

as resources presented in this paper) for clients to read and discuss. If this information is provided, the subsequent discussion should be centered around client self-determination, and the therapist should also partner with the client by using the terms the clients use to conceptualize the practices. What a client chooses to call a religious-based mindfulness practice is less important than the purpose and benefits being sought.

Finally, it can be important for therapists to establish relationships with a range of religious leaders for professional consultation over a number of mental health topics that intersect with religious beliefs. These professional relationships could also be used by therapists unfamiliar with Christian traditions to better understand religious practices and rituals. This seeking out of religious consultation should be done carefully and with attention to the vast differences in denominational interpretations. As such, the consultation should be more about general learning and understanding for the therapist and not about how the therapist should direct the client in their tradition. If a client wants to seek their own consultation, perhaps a referral to a trained spiritual director or counselor in their tradition would be appropriate.

These suggestions above seek to support therapists in scaffolding Christian clients who already use or want to use intentional religious coping. It also hopes to assuage concerns that mindfulness practices are in conflict with Christianity. Although not all Christians need theological connections and adaptions for mindfulness practice, there is a continued benefit to bring Christian traditions into deeper conversation with mindfulness practices. Clinicians can present scientific evidence and resources, such as those noted in this paper, to help clients adapt mindfulness practices and incorporate their own spiritual and worldviews. Christian leaders can also benefit from learning more about the overlaps of mindfulness and Christianity in order to support religious growth and coping, without fear based in misunderstandings of mindfulness.

**Author Contributions:** Conceptualization, V.L.T. and J.C.H.; resources, V.L.T.; writing—original draft preparation, V.L.T.; writing—review and editing, V.L.T. and J.C.H.; All authors have read and agreed to the published version of the manuscript.

**Funding:** This research received no external funding.

**Institutional Review Board Statement:** Not applicable.

**Informed Consent Statement:** Not applicable.

**Data Availability Statement:** Not applicable.

**Conflicts of Interest:** The authors have no conflicts of interest to disclose.

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
