# Peer review of "Christian Mindfulness and Mental Health: Coping through Sacred Traditions and Embodied Awareness"

_religions, doi:10.3390/rel13010062_

Round 1

Reviewer 1 Report

Basic argument: The authors are arguing that, in cases where a Christian is seeking support from a meditative practice, it can be appropriate to recommend a Christian practice (I am not convinced by this, but the thesis is fine) due to similar effects. Second, that there is a need to adapt Christian theologies to somatic experiences that mindfulness works with, so that their use of mindfulness practices can be ‘Christian’. Again, I’m fine with that (but it would be a fair bit of work). The main element I am not totally comfortable with is how the authors identify the equivalencies between the Christian and mindfulness practices (sections cited below). Nevertheless, since the empirical work on the Christian practices seem to provide similar benefits to the mindfulness practices, I don’t think the authors have over-reached. I would ask them to consider more closely the problem of the conditions of the convergences. Are these due to the mechanisms of the practices or the objects or neither or both? It would benefit the paper to show that the mechanisms of Christian meditations have resonance with the processes and behaviours that constitute mindfulness. This connexion would provide some basis for a claim that Christian practices and mindfulness producing similar effects. This intersection seems quite key to me. I would ask the authors spell to this out a little more, after which I am confident the paper will be ready.

50-57. I think that the intentional awareness definition (Mindfulness def) might be too generic, as, in practice, it is accompanied by specific, mostly somatic, foci. This means the resemblance between Christian and mindfulness behaviours cannot be isomorphic (at least not without a clear argument) because something like God or spirit as objects of intentional awareness are very different from somatic foc. Intentional awareness of these things may depend on conceptual/ interpretive/ discursive principles, rather than a non-conceptual or somatic object. What I mean is that effects of mindfulness are not determined only by behaviours, but also by the objects of attentional/ intentional awareness, so I am not convinced that an argument can be made about the equivalence of effect when the objects of attention vary so much.  

173-192: I’d like the authors to clarify some of the differences between the operations occurring within the Christian and mindfulness practices. To me, there seems sufficient similarity between the processes that constitute Christian and mindfulness practices to identify operational similarities between the two. I think it would be helpful to clarify the relationship between behaviours and objects. Are the behaviours (internal processes) used by the Christian during a Christian practice, are equivalent or dissimilar to the mindfulness will also depend on the character of the attentional object.  Is it appropriate to identify the operational model of mindfulness (attention and open monitoring) with single-pointed attention as the use of the Jesus prayer, and something approximating open-monitoring and God? Despite the identification of attention with God, the object, God, is often dispersed, generic, and non-somatic, but also non-judgemental, loving etc. To me, this suggests that the character of the objects is qualitatively distinct as to almost swap open-monitoring with attention, or even combining the behaviours of attention and open-monitoring into one. However, as you suggest later, it might be quite appropriate to have somatic basis for the experience of God, in that case, then, perhaps having the object ‘God’ can be quite concrete and specific, although it might be worth clarifying this.

Editor’s personal notes: Perhaps one of the reasons that Christian and mindfulness practices share some similar positive effects is because elements of the practices approximate the other. Perhaps a Christian meditation has a unique interplay between attention focus and OM, in which the object (God) provides the formative attitudes rather than the subject (as it is for Buddhist-based approaches). The reason I mention this is because the later engagement with the body will tend, in the practices, alternate between OM and attention in various ways.

Author Response

Reviewer 1

Point 1: 

The authors are arguing that, in cases where a Christian is seeking support from a meditative practice, it can be appropriate to recommend a Christian practice (I am not convinced by this, but the thesis is fine) due to similar effects. Second, that there is a need to adapt Christian theologies to somatic experiences that mindfulness works with, so that their use of mindfulness practices can be ‘Christian’. Again, I’m fine with that (but it would be a fair bit of work). The main element I am not totally comfortable with is how the authors identify the equivalencies between the Christian and mindfulness practices (sections cited below). Nevertheless, since the empirical work on the Christian practices seem to provide similar benefits to the mindfulness practices, I don’t think the authors have over-reached. I would ask them to consider more closely the problem of the conditions of the convergences. Are these due to the mechanisms of the practices or the objects or neither or both? It would benefit the paper to show that the mechanisms of Christian meditations have resonance with the processes and behaviours that constitute mindfulness. This connexion would provide some basis for a claim that Christian practices and mindfulness producing similar effects. This intersection seems quite key to me. I would ask the authors spell to this out a little more, after which I am confident the paper will be ready.

 Response 1: I appreciate this note on needed clarification. To better identify the convergences and divergences of Christian practices and mindfulness practices, I added a section (now section 2 on page 2) on some of the general assumptions of Buddhism and Christian concepts of mindfulness. Specifically, I bring up the similarities in practices related to decreasing selfishness and increases connection to something beyond once self; however, I also point out how in Christian practices these are directed at the object of God versus the non-theistic and detachment emphasis in Buddhism. This change also addressed a comment by reviewer 2 and strengthened the paper overall.

Point 2:

50-57. I think that the intentional awareness definition (Mindfulness def) might be too generic, as, in practice, it is accompanied by specific, mostly somatic, foci. This means the resemblance between Christian and mindfulness behaviours cannot be isomorphic (at least not without a clear argument) because something like God or spirit as objects of intentional awareness are very different from somatic foc. Intentional awareness of these things may depend on conceptual/ interpretive/ discursive principles, rather than a non-conceptual or somatic object. What I mean is that effects of mindfulness are not determined only by behaviours, but also by the objects of attentional/ intentional awareness, so I am not convinced that an argument can be made about the equivalence of effect when the objects of attention vary so much.  

Response 2: I added a few lines that connect liturgical practices and rituals with intentional awareness and elements of mindfulness more explicitly. With the addition of these statements and section 2 as briefly summarized above, I feel the reviewer’s comment has been addressed. It seems that there is a both/and aspect to the connections and divergences between mindfulness practices and specific Christian adaptions of these practices that I hope is clearer with the additional historical and conceptual information.

173-192: I’d like the authors to clarify some of the differences between the operations occurring within the Christian and mindfulness practices. To me, there seems sufficient similarity between the processes that constitute Christian and mindfulness practices to identify operational similarities between the two. I think it would be helpful to clarify the relationship between behaviours and objects. Are the behaviours (internal processes) used by the Christian during a Christian practice, are equivalent or dissimilar to the mindfulness will also depend on the character of the attentional object.  Is it appropriate to identify the operational model of mindfulness (attention and open monitoring) with single-pointed attention as the use of the Jesus prayer, and something approximating open-monitoring and God? Despite the identification of attention with God, the object, God, is often dispersed, generic, and non-somatic, but also non-judgemental, loving etc. To me, this suggests that the character of the objects is qualitatively distinct as to almost swap open-monitoring with attention, or even combining the behaviours of attention and open-monitoring into one. However, as you suggest later, it might be quite appropriate to have somatic basis for the experience of God, in that case, then, perhaps having the object ‘God’ can be quite concrete and specific, although it might be worth clarifying this.

Response 3: These are great notes and questions. I hope the additions I made in section 2 and in the conclusion have addressed this comment as best as possible given the length and scope of the paper.

Reviewer 2 Report

Thank you for the opportunity to review your manuscript on Christian mindfulness and mental health. This is an important topic, and one that I have encountered in my own clinical work and research. The authors do a good job of explaining how mindfulness is rooted in Christian theology and how this can be used to assuage the concerns of Christian clients. I have just a few suggestions that could further strengthen this work.

-I think it would be helpful at the beginning to say more about the Buddhist roots of mindfulness, since this is the main concern referenced by some Christian clients. This might happen in the first paragraph on page 2 where you briefly mention Buddhism. Outlining in more detail the way mindfulness is usually taught and its roots in Buddhism will provide the context necessary for then showing roots in Christianity.

-Similarly, I wasn't clear if mindfulness was first described/practiced in Christianity or in Buddhism or if it occurred in tandem. Perhaps it is a "both-and" situation, and if so, I'd state this so that the argument for Christian use/adaptation is clearer. Also, if mindfulness is rooted in Christianity, how did we end up only being taught about its Buddhist roots? Presenting this information will also help the concerned client (and confused therapist).

-For section 2.2 on page 3, please cite the studies that you are referring to when you say that mindfulness adopted for Christian beliefs is an emerging area of study.

-You do a very nice job with your theological explanations. I was hoping for a little more in terms of concrete application, especially for a non Christian (or a non informed therapist regarding Christianity). Can you add more about what a therapist can do who was taught the Buddhist roots of mindfulness? What if the therapist is not familiar with this theology or types of prayer or rituals or sacraments? What is this therapist to do? Do they teach the same mindfulness exercises but provide more Christian explanation? More clarity and concrete suggestions would be helpful.

Author Response

Reviewer 2

Point 1: Thank you for the opportunity to review your manuscript on Christian mindfulness and mental health. This is an important topic, and one that I have encountered in my own clinical work and research. The authors do a good job of explaining how mindfulness is rooted in Christian theology and how this can be used to assuage the concerns of Christian clients. I have just a few suggestions that could further strengthen this work.

Response 1: Thank you for this feedback and perspective!

Point 2: I think it would be helpful at the beginning to say more about the Buddhist roots of mindfulness, since this is the main concern referenced by some Christian clients. This might happen in the first paragraph on page 2 where you briefly mention Buddhism. Outlining in more detail the way mindfulness is usually taught and its roots in Buddhism will provide the context necessary for then showing roots in Christianity.

Response 2: I have added a section for this now on page 2. I feel this greatly strengthen the paper and clarified some of the difference and similarities in the assumptions of mindfulness.

Point 3: Similarly, I wasn't clear if mindfulness was first described/practiced in Christianity or in Buddhism or if it occurred in tandem. Perhaps it is a "both-and" situation, and if so, I'd state this so that the argument for Christian use/adaptation is clearer. Also, if mindfulness is rooted in Christianity, how did we end up only being taught about its Buddhist roots? Presenting this information will also help the concerned client (and confused therapist).

Response 3: It is hard to know who “got there first” I did do my best to note that these practices were both active in the first millennium BC in Judaism/foundation of Christianity and Buddhism as well as the precursors to Buddhism. I hope to have clarified this somewhat given the space and scope of the paper. It seems mindfulness is one of the ancient practices for many people groups.  

Point 4: For section 2.2 on page 3, please cite the studies that you are referring to when you say that mindfulness adopted for Christian beliefs is an emerging area of study.

Response 4: These citations have been added.

Point 5: You do a very nice job with your theological explanations. I was hoping for a little more in terms of concrete application, especially for a non Christian (or a non informed therapist regarding Christianity). Can you add more about what a therapist can do who was taught the Buddhist roots of mindfulness? What if the therapist is not familiar with this theology or types of prayer or rituals or sacraments? What is this therapist to do? Do they teach the same mindfulness exercises but provide more Christian explanation? More clarity and concrete suggestions would be helpful.

Response 5: Thank you for this nudge to include more specific suggestions. I have added 2 paragraphs to the conclusion and implications section to provide some suggestions. I feel this addition captures the stated goals of the paper better.
